# Role of Endothelial Progenitor Cells in Frailty

**DOI:** 10.3390/ijms24032139

**Published:** 2023-01-21

**Authors:** Klara Komici, Angelica Perna, Germano Guerra

**Affiliations:** Department of Medicine and Health Sciences, University of Molise, 86100 Campobasso, Italy

**Keywords:** endothelial progenitor cells, frailty, aging, endothelial dysfunction

## Abstract

Frailty is a clinical condition closely related to aging which is characterized by a multidimensional decline in biological reserves, a failure of physiological mechanisms and vulnerability to minor stressors. Chronic inflammation, the impairment of endothelial function, age-related endocrine system modifications and immunosenescence are important mechanisms in the pathophysiology of frailty. Endothelial progenitor cells (EPCs) are considered important contributors of the endothelium homeostasis and turn-over. In the elderly, EPCs are impaired in terms of function, number and survival. In addition, the modification of EPCs’ level and function has been widely demonstrated in atherosclerosis, hypertension and diabetes mellitus, which are the most common age-related diseases. The purpose of this review is to illustrate the role of EPCs in frailty. Initially, we describe the endothelial dysfunction in frailty, the response of EPCs to the endothelial dysfunction associated with frailty and, finally, interventions which may restore the EPCs expression and function in frail people.

## 1. Introduction

Population aging is a global phenomenon, and the number of persons aged 65 years or over reached 727 million in 2020 [1]. By 2050, the number of older persons is expected to double, and the proportion of the elderly is projected to increase by up to 16% such that one in six persons will be over the age of 65 [2]. 

Frailty is a clinical condition closely related to ageing which is characterized by a multidimensional decline in biological reserves, a failure of physiological mechanisms and vulnerability to minor stressors [3,4]. Frailty is associated with an increased risk of adverse outcomes including disability, falls, delirium and mortality [5,6,7,8]. The clinical presentation of frailty is characterized by fatigue, unexplained weight loss, frequent infections, balance and gait impairment, impaired awareness and fluctuating disability. Demographic and social factors, physical performance, impaired cognition, multiple chronic diseases and malnutrition are considered crucial risk factors for the onset and progression of frailty [9]. 

Chronic inflammation, the impairment of endothelial function, the age-related modification of the endocrine system and immunosenescence cross-linked to genetic and environmental factors are important mechanisms in the pathophysiology of frailty. Several inflammatory cytokines including IL-6, tumor necrosis factor-α (TNFα) and the acute phase protein C-reactive protein (CRP) have been independently associated with frailty [10]. Inflammation is associated with the catabolism of skeletal muscles and a reduction in muscle strength, which characterize frailty [11]. The alteration of insulin growth factor (IGF-1) signaling, the reduction in testosterone and estradiol levels and the modification of cortisol release influence the development of vulnerability and frailty [12]. Furthermore, the blunting of the B-cell-controlled antibody response, changes in T-lymphocyte production, the impaired activity of macrophages and a reduction in the number of stem cells respond inappropriately to inflammation, injury and endothelial damage [13]. 

Endothelium represents a functional barrier between tissues and circulating blood, prevents platelet and leukocyte aggregation and adhesion and produces a variety of vasoregulation factors such as nitric oxide (NO) and endothelins [14]. Alteration in the endothelial cells (ECs) damage/repair balance leads to vascular remodeling and the impairment of NO availability and is a key element in endothelial dysfunction and the atherosclerosis process [14,15]. Atherosclerosis is considered an age-dependent disease and leads to the development of cardiovascular comorbidities [15]. Furthermore, endothelial dysfunction has been suggested as an early predictor of frailty phenotype onset [16]. 

Endothelial progenitor cells (EPCs) are considered important contributors in the endothelium homeostasis and turn-over. They replace dysfunctional ECs, facilitate re-endothelization and influence the process of age-related vascular remodeling [17]. In the elderly, EPCs are impaired in terms of function, number and survival, and the modification of the circulating EPCs level in relation to physical frailty has been reported [18]. In addition, the modification of the EPCs level and function has been widely demonstrated in atherosclerosis, hypertension, diabetes mellitus, rheumatic disorders and dementia, which are the most common age-related diseases [19,20,21,22,23]. 

Considering the age-related modification of EPCs, the effect of multiple comorbidities and chronic conditions on EPCs and endothelial dysfunction, EPCs may have an important role in the underlying mechanisms of frailty. In addition, the negative impact of frailty is also characterized by a low quality of life for older people, the failure of the therapeutic management of chronic conditions and increased healthcare costs. Interventions guided by comprehensive geriatric assessments (CGA), which include nutritional support, physical activity and the revision of drug therapy, are beneficial for frail people, but no specific therapy for frailty exists. The purpose of this review is to illustrate the role of EPCs in frailty. Initially, we describe the endothelial dysfunction in frailty, the response of EPCs to the endothelial dysfunction associated with frailty and, finally, interventions that may restore the EPCs expression and function in frail people.

### 1.1. Main Characteristics of EPCs

EPCs are a heterogeneous group of cells of different origins and in different stages of maturation which contribute to endothelial regeneration and vascular repair [24,25]. The mobilization of EPCs from bone marrow to circulation is controlled by a variety of angiogenic factors, including endothelial NO synthetase (eNOS) and vascular endothelial growth factor (VEGF), which play an important role by enhancing the growth of EPCs, tube formation and the angiogenesis process [26,27]. 

Based on the phenotype and biological functions of EPCs, two distinct subtypes have been proposed: endothelial colony-forming cells (ECFCs) and myeloid angiogenic cells (MACs) [28]. 

ECFCs derive from umbilical cord blood or peripheral blood mononuclear cells and are characterized by CD31+, CD45, CD14, CD146+, VE-Cadherin+, von Willebrand factor+ and VEGFR2+ phenotype. Their vasculogenic properties are linked to platelet-derived growth factor BB (PDGF-BB)/platelet-derived growth factor receptor (PDGFR) signaling [29,30]. 

MACs derive from peripheral blood mononuclear cells grown under endothelial cell culture conditions and are characterized by the following surface cell markers: CD45+, CD14+, CD31+, CD146, CD133 and Tie2. MACs promote angiogenesis through the activation of IL-8/VEGFR2/ERK signaling pathways, which results in endothelial proliferation, migration and tube formation. MACs do not differentiate into ECs but enhance the migration of circulating or vascular wall ECFCs to the injury area, where ECFCs proliferate, differentiate into mature ECs and restore the endothelial integrity of the vascular wall [25].

EPCs homing in the site of injury is characterized by recruitment, mobilization, adhesion and CXCR2, CXCR4 and CCR2 signaling. EPCs’ functional capacity has been reported to ameliorate after CXCR4 increased the expression via AKT/endothelial NO synthase pathways [31], and the augmentation of integrin receptor subunits present in EPCs promotes adhesion properties [32].

Endothelial injury is characterized by the up-regulation of intracellular and vascular adhesion molecules (ICAM-1) (VCAM-1) and hypoxia-inducible factor-1a (HIF-1a), which regulates the release of stromal-derived factor-I alfa (SDF-1a) and VEGF. These factors mediate the trafficking and recruitment of MACs to the target tissue [33,34]. MACs do not directly supply new ECs but activate the resident ECs through the release of growth factors, cytokines and transcription factors [35,36]. The paracrine release of VEGF, hepatic growth factor (HGF), Ang-1, SDF-1a, insulin-like growth factor (IGF)-1 and eNOS by MACs recruits and incorporates ECFCs to the network of new capillary vessels [37,38]. In addition, ECFCs release pro-angiogenic factors [39] and regulate the regenerative potential of mesenchymal stem cells [29]. Despite the exact origin of ECFCs not being clear, the identification of the endothelial progenitor/stem like population at the inner surface of the pre-existing blood vessels with colony-forming abilities has been reported [40]. ECFCs are crucial for the vascular repair and endothelial regeneration in different organs. Indeed, ECFCs have been shown to promote neovascularization and increase microvascular density in myocardial ischemia [41], ischemic stroke [42] and hindlimb ischemia [43]. 

Chronic inflammation and the release of inflammatory cytokines lead to the excessive proliferation of smooth muscle cells (SMCs), extracellular-matrix (ECM) deposition and the trans-differentiation of SMCs in myofibroblasts (MFs) [44]. MFs promote the activation of pro-inflammatory angiogenic factors and the modification of ECM. The excessive production of IL-6 and VEGF may negatively affect EPCs’ mobilization and recruitment, leading to an overactivation and defective vascular remodeling [17]. 

Increasing age was associated with lower circulating EPCs and lower VEGF levels in patients undergoing coronary artery bypass grafting. Notably, by-pass surgery could not provide a significant mobilization in EPCs in patients aged 69 years and older [45]. 

Another study including heart failure with preserved ejection fraction reported that the EPCs of older patients with heart failure with mild reduced ejection fraction presented lower migratory, proliferative and adhesion properties [46]. Compared to younger patients with NSTEMI myocardial infarction, the EPCs presented impaired function, and this was associated with the severity of the disease [47]. 

A significant reduction in CXCR4 expression in EPCs with aging has been described, and the modification of calcium-regulated functions of the CXCR4/ SDF-1a axis has been suggested as a possible mechanism [48,49]. Notably, the upregulation of CXCR4/JAK-2 signaling has been associated with the amelioration of EPCs properties and the restoration of endothelial dysfunction related to age [50]. 

#### Isolation, Culturing and Measurement of Human EPCs

A variety of methods are used for the isolation and quantification of EPCs from peripheral blood. However, cell surface phenotyping and cell cultures have been proposed as the main approaches [28,51,52].

Cell surface phenotyping is based on flow cytometry and fluorescent labeled antibodies. Flow cytometry requires a small amount of blood, and circulating EPCs are quantified as the percentage of the mononuclear cells that express VEGFR2 and CD34 [53]. In addition, CD133 surface protein has been associated with properties of EPCs by different studies [54,55]. Based on these studies, EPCs were identified as circulating CD34 cells that co-express VEGFR2 and CD133 as well. Furthermore, the quantification of EPCs by this approach could provide information regarding the relationship between the number of EPCs and the different diseases state. It should be mentioned that CD34, CD133 and VEGFR2 are not unique for EPCs, and their expression is also present in hematopoietic cells [56]. Other studies have reported that CD34 and CD45 cells enriched for hematopoietic cells co-expressed CD133 without VEGFR2. In addition, the CD34CD45 population of cells, which gave rise to colonies of ECs with high proliferative properties, expressed VEGFR2 without CD133 [51,57].

The isolation of EPCs by the cell culture approach was initially described by Asahara and colleagues [58]. Peripheral blood mononuclear cells were plated on fibronectin-coated dishes within five days, and the adherend cells expressed similar cell surface proteins to human umbilical vein ECs. To deplete the population of macrophages and monocytes that could contaminate the isolation of EPCs, Ito and co-workers [59] after a twenty four-hour period of adhesion, removed the non-adherent cells and re-plated them onto fibronectin-coated dishes for seven days. The cells that emerged after one week of the culture were considered as EPCs colonies. Another study cultured mononuclear cells from peripheral blood for forty-eight hours, and the non-adherent fraction was re-plated in a specific medium onto fibronectin-treated petri dishes. After five to seven days, colony-forming units (ECs) were obtained and named as CFU-Hill [60]. However, other studies demonstrated that CFU-Hill colonies include monocytes, myeloid progenitor cells and T lymphocytes [61,62]. 

The adherent cells obtained from circulating mononuclear cells plated in culture dishes in an endothelial growth medium present a morphology resembling the spindle-shape cells similar to CFU- Hill after three to four days of the culture and express markers typical of the ECs, such as von Willebrand factor VE-cadherin, CD31 and Tie2. Furthermore, these cells were demonstrated to promote vascularization [63] and are identified as early EPCs or circulating angiogenic cells (CACs). Considering that these cells are generated in in vitro conditions, it has been suggested to refer to them as MACs [28]. 

Late-outgrowth EPCs are isolated from the mononuclear cells of peripheral blood or umbilical cord blood adherent to collagen-I-coated culture dishes after two weeks of the culture. These cells are defined as ECFCs, differentiate into mature ECs and generate new vessels in vivo [64]. 

### 1.2. Frailty Models and Epidemiology

Several instruments have been developed to detect frailty, and, currently, the two main frailty models are: (a) the Fried Frailty Phenotype and (b) the Frailty Index (FI) based on the cumulative deficit model. The frailty phenotype model was established by the presence of the following criteria: (a) unintentional weight loss, (b) self-reported exhaustion, (c) low energy expenditure, (d) slow gait speed and (e) weak grip strength. The presence of one to two of these criteria identifies a pre-frail condition, while people with three or more criteria are considered frail. Despite multiple comorbidities and the presence of cognitive decline not being included in the development of this model, the application of the Frailty Phenotype has been validated, and the identification of frailty with this model has an independent role in the prediction of long-term mortality [7,65]. 

The cumulative deficit model is computed by the number of health deficits identified in different symptoms, signs, laboratory findings, comorbidities and disabilities [66]. The frailty index is derived from the number of health deficits divided by the total number of variables screened. Importantly, a value of 0.67 seems to identify an amount of frailty that is highly associated with mortality [66]. Several studies have reported that the Frailty Index was strongly related to institutionalization and poor survival [67,68]. 

Based on these models, many other screening tools have been developed and validated in different populations and clinical settings [69]. For instance, evaluations of the level of deficits based on CGA have been widely applied in different clinical settings [70,71,72]. The Study of Osteoporotic Fractures (SOF) frailty scale is a parsimonious frailty index including: weight loss, the inability to rise from a chair and poor energy, and it is able to predict disability, fractures and falls [73]. The Tilburg Frailty Indicator and the Groningen frailty indicator are questionnaires which evaluate physical, cognitive, social and psychological domains [74]. It should be mentioned that, in terms of the ability to predict negative outcomes, frailty models and screening tools demonstrate a degree of overlap, but different patient cohorts are identified as frail. 

The prevalence of frailty varies by classification, sex and geographic region; however, an overall pooled prevalence of 12% has been estimated considering the Frailty Phenotype model or other tools focused on physical frailty [75]. The cumulative deficit model of frailty produces higher estimates of population-level prevalence, accounting for about 24% [75]. For pre-frailty, the overall estimates are 46% for physical frailty and 49% for cumulative deficit models, respectively. The prevalence increases with age, and higher proportions of frailty and pre-frailty are registered among the female population [76]. Regarding prevalence by geographic areas, in Europe, it is 8%, and the highest prevalence is in Africa (about 22%), considering the physical frailty model. Frailty evaluation by the cumulative deficit model produces the highest prevalence in Oceania (31%), followed by Asia (25%). In Europe, this model estimates an overall prevalence of frailty of 22%. 

## 2. Impairment of Endothelial Function in Frailty

Endothelium function includes the exchange of molecules and fluids between blood and surrounding tissues, the maintenance of blood in a fluid state, the facilitation of the immune response, the control of vascular resistance, the regulation of the vascular tone and the creation of a new vascular network [14,17]. The endothelium acts through the paracrine and endocrine pathways and constantly maintains an equilibrium between vasodilatation and vasoconstriction, pro-inflammatory and anti-inflammatory mediators and pro-thrombotic and antithrombotic factors. The impairment of endothelial function is characterized by pro-inflammatory, prothrombotic and vasoconstrictor properties. A blunted response to agonist-induced vasodilation, the impairment of synthesis and the release of NO are the main characteristics of the dysfunctional endothelium. 

Data from the Cardiovascular Health Study have reported that frail people are characterized by increased intima-medial thickness of the carotid arterial wall and a reduction in blood flow in the lower extremities despite the absence of clinical cardiovascular diseases [77]. For the first time, the Toledo Study for Healthy Aging revealed an association between endothelial dysfunction and frailty [78]. In this study, pre-frail and frail subjects were characterized by a higher asymmetric dimethylarginine (ADMA) level, which is an endogenous inhibitor of NO synthase. In contrast, the ADMA level was not significantly different in frail and/or pre-frail people compared to that in non-frail people, where atherosclerosis was present. In people without cardiovascular diseases, a higher ADMA level significantly increases the risk of frailty independently by well-established cardiovascular risk factors such as: hypertension, diabetes and dyslipidemia. Chronic inflammation and oxidative stress were suggested as complementary sources of endothelial dysfunction in frailty [79]. Another study also found that serum ADMA levels were correlated with physical domains of frailty. Higher ADMA levels are significantly associated with a lower muscle mass and muscle strength and a slower gait speed [80].

A cross-sectional study investigating the impact of different inflammatory mediators on the frailty status in elderly outpatients found that plasma nitrite levels, which are mainly derived from constitutive NO synthetase activity, were significantly reduced in frail people. This finding was linked to the presence of chronic low-grade inflammation, resulting in progressively increased CRP in frail people. However, after the adjustment for confounders, CRP was not significantly associated with frailty, whereas nitrite levels showed an independent role [81].

Brachial artery flow-mediated dilatation (FMD) and brachial angle pulse wave velocity (baPWV) are indirect measures of endothelial function, and their modifications indicate endothelial impairment, which is associated with atherosclerosis risk. FMD was significantly associated with a lower limb muscle power [82] and muscle mass index in the elderly population [83]. Furthermore, baPWV was negatively associated with handgrip strength dominantly among the non-hypertensive population [84]. Another study including thirty hospitalized elderly patients, where the frailty status was evaluated by the SOF frailty scale, revealed a lower FDD in frail compared to non-frail patients [85]. Frailty was also associated with an abnormal FDD in elderly patients with chronic kidney disease [86]. A lower ankle-brachial index was associated with frailty in two studies, but the analysis adjusted for confounders failed to show a significant correlation [87,88]. A recent cross-sectional analysis of 1096 men revealed that the femoral angle PWV was significantly associated with frailty among men without cardiovascular diseases but not in men with cardiovascular diseases [89]. 

Interestingly, a recent experimental study observed that the ex vivo measurement of endothelial-dependent dilatation (EDD) and the superoxide-mediated suppression of EDD were impaired in frail mice, independently of age [90]. Mouse frailty was determined by a 31-item frailty index based on clinical signs of deterioration in C57BL/6J mice [91]. The evaluation of the vestibulocochlear, ocular, nasal, respiratory, musculoskeletal, digestive and urogenital signs was included. Notably, endothelial dysfunction was significantly associated with frailty among old female and male mice, but the localization was cerebral arteries for female mice and the mesenteric artery for male mice. 

## 3. EPCs Response to Endothelial Dysfunction in Frailty 

The plasma concentration of ADMA has been negatively correlated with the number of MACs and ECFCs. The in vitro differentiation of EPCs was repressed in a concentration-dependent manner by ADMA, which also significantly reduced the incorporation of EPCs into endothelial tube-like structures [92]. Furthermore, the formation of colony-forming units from cultured peripheral blood mononuclear cells was inhibited. In this study, the detrimental effects of ADMA on EPCs were abolished by the co-incubation with the hydroxymethyl glutaryl coenzyme A reductase inhibitor [92]. Other studies have confirmed the suppression of EPCs by ADMA [93,94,95]. The dimethylarginine dimethylaminohydrolase (DDAH2)/ADMA pathway, through the activation of silent inhibitor 1, accelerated the senescence of EPCs [96]. ADMA’s negative role in EPCs function was also explained by the endoplasmatic reticulum (ER) stress pathway through the activation of phosphorylated protein kinase RNA-activated-like ER kinase (PERK), a stress sensor protein. In addition, the inhibition of the ER stress pathway by salubrinal attenuated the ADMA-induced apoptosis of EPCs [97]. Another in vitro study found that ADMA promoted EPCs apoptosis through the phosphorylation of JNK, targeted the inhibition of the JNK by SP600125, alleviated ADMA-induced apoptosis and promoted angiogenesis viability [98].

It has been reported that NOS inhibition attenuated the migration properties of EPCs, while NO donors enhanced VEGF-dependent chemotaxis. Importantly, eNOS levels were also significantly reduced in older patients as compared to healthy volunteers [99]. It has also been demonstrated that eNOS is present in EPCs and is dynamically expressed during the differentiation of EPCs to ECs [100].

The reduction in VEGF expression with aging has been described [101], and VEGF promoted EPCs incorporation into the damaged vessels, enhanced the differentiation of EPCs, as indicated by the increased expression of the EC markers CD31 and vWF, and promoted re-endothelization. The possible mechanism related to the VEGF-induced modification of EPCs has been mediated by connexin 43(Cx43), which is a gap junction protein and an important contributor in the intracellular communication. Indeed, VEGF increases the expression of Cx43 in EPCs and the inhibition of Cx43 expression using short interfering RNA (siRNA) attenuated EPCs gap junction intercellular communication and consequent EPCs differentiation [102]. 

Interestingly, a recent study revealed that VEGF signaling was greatly reduced in multiple key organs in an experimental model of mouse aging [103]. This was associated with the increased production of soluble VEGFR1, produced through an age-related shift in the alternative splicing of VEGFR1 mRNA, and its activity as a VEGF trap. VEGF supplementation resulted in increased longevity in mice. Considering the strict correlation between VEGF, NOS and EPCs, a reduction in VEGF and NOS expression or an impairment in their signaling is paralleled by a reduction in the impairment of the circulating EPCs number and by the impairment of their differentiation. The role of VEGF and NO in vascular aging has been studied widely; however, more research is necessary to clarify their role in frailty models and, consequently, the modification of EPCs. It has been suggested that the exchange of cAMP/PKA in gap junction proteins could interfere with EPCs functioning, and the blockade of PKA attenuates VEGF-induced EPCs differentiation [102,104]. It has been reported that VEGF activates phospholipase C and phosphatidylinositol-4,5-bisphosphate, which results in a cellular calcium increase and the regulation of endothelial differentiation-related genes. 

The main mechanisms responsible for EPCs modification in frailty are represented in Figure 1.

## 4. EPCs Dysfunction and Frailty Domains

It has been demonstrated that lower circulating EPCs are associated with a lower gait speed, a lower six-minute walk distance, a longer time for the chair stand test and a reduced SF-36 physical function score. In this study, EPCs were analyzed by flow cytometry and the analysis of cell surface expression markers: CD34+, CD133+, CD14+ and CD146+. It should be mentioned that the association of EPCs and physical function remained significant across several similar immature EPCs and multiple measures of physical function adjusted for age, BMI, comorbidities and markers of inflammation [105].

The same authors revealed that circulating EPCs levels are not only associated with baseline physical performance but are also able to predict the physical function decline. Lower baseline levels of EPCs were highly predictive of a lower gait speed and a shorter distance walked in 6 min at 3 and 12 months of follow-up. These associations were significant after the adjustment for age, body mass index and inflammation and were independent of interventions provided to improve physical function [106]. 

In addition, a cross-sectional study including community-dwelling elderly people reported a positive association between circulating EPCs and handgrip strength in hypertensive men. This association remained robust after the adjustment for cardiovascular risk factors [18]. 

Regarding cognitive function, which is another crucial domain of frailty, it has been reported that a lower level of EPCs was associated with the presence of mild cognitive impairment in the elderly. In addition, a lower circulating level of EPCs was present in patients with worse verbal and visual memory [107]. In contrast, a later study found no association between the EPCs level and cognitive decline in elderly subjects, suggesting that the multimorbidity observed in our patients may lead to opposite and confounding effects on endothelial biomarkers levels [108]. However, another study performed on 509 patients showed that lower counts of EPCs were associated with a worse memory performance and cognitive impairment in patients with coronary artery disease, and the association of EPCs with visual and verbal memory remained significant even after adjusting for confounders [109]. 

It has been postulated that coronary artery disease, hypertension, diabetes and aortic stenosis are characterized by the impairment of EPCs, and the association between cardiovascular comorbidities and frailty is well established. In addition, the EPCs number and migration were significantly decreased in patients with COPD, another important comorbidity related to frailty [110,111]. Notably, in community-dwelling elderly people affected by osteoporosis, a significant association between the circulating osteogenic cells population, disability and frailty based on the Fried criteria and the deficit model was identified [112]. Recent studies have underlined the potential relationship between reduced levels of circulating CD34+ hemopoietic progenitor stem cells (HSPCs) and frailty [113,114]. Low HSPCs are associated with pre-frailty and frailty and identify individuals with poor cardiovascular outcomes [114]. 

It should be mentioned that few studies have investigated the association of EPCs with frailty domains, and, moreover, frailty models/EPCs relationships should be explored.

## 5. Interventions That May Restore EPCs Functioning in Frailty

The effects of physical activity on frailty trajectories, mobility and disability are well documented in different studies [115,116,117,118,119]. The severity of frailty was reduced through one year of supervised home-based physical training [120]. A recent interventional study including patients with chronic heart failure reported that an exercise rehabilitation program enhanced the proliferation, migration and activity of EPCs. At the same time, the apoptosis rate was lower compared to that of the control group. Furthermore, the mRNA expression of PI3K, AKT, eNOS and VEGF was significantly higher in the intervention group compared to that in the control group [121]. 

Angiotensin II may promote the impairment of endothelial function and has adverse effects on skeletal muscle function and structure in experimental models [122], and elderly women who had taken ACE inhibitors continuously presented a lower decline in muscle strength compared with those who had taken other antihypertensive drugs or those who had never used antihypertensives during a long-term follow-up [123]. It has been reported that ACE inhibitors increased the EPCs level and restored the function of EPCs in hypertensive patients [124]. However, the existing evidence does not support the use of ACE inhibitors or angiotensin receptor blockers as a single intervention for improving physical performance in the elderly [125], and a recent study did not find an improvement in muscle mass related to perindopril and leucin supplementation [126].

A recent study demonstrated that, in patients with heart failure (HF) and diabetes, dapagliflozin, a sodium-glucose cotransporter (SGLT2) inhibitor, provided beneficial effects related to the worsening of HF, hospitalizations and cardiovascular death, regardless of frailty class. Furthermore, in patients with a greater degree of frailty, improvements in symptoms, physical function and quality of life were larger [127]. Even if previous research suggested that SGLT2 inhibitors did not influence the number of EPCs [128], a recent study revealed that canagliflozin increased the CXCR4 receptor expression and migratory profile of EPCs in patients with diabetes [129]. Furthermore, recent evidence suggests that empagliflozin ameliorates the frailty status in elderly people with diabetes by improving endothelial function via the reduction of mitochondrial oxidative stress [130].

## 6. Conclusions

The current evidence suggests that a reduction in the EPCs number characterizes frailty. The impairment of EPCs function may be a possible independent mechanism involved in the development of frailty. Increased ADMA expression, a reduction in VEGF and an impairment of NOS signaling appear to be the main pathways related to EPCs dysfunction in frailty. EPCs may represent a potential biomarker in the early detection of pre-frailty and frailty, the progression of frailty and the monitoring of interventions guided by frailty deficits. However, the role of EPCs as biomarkers capable of identifying frailty, the impact of EPCs in the clinical outcome of frail people and the modification of EPCs secondary to strategies which aim to ameliorate frailty should be further investigated. 

## Figures and Tables

**Figure 1 ijms-24-02139-f001:**
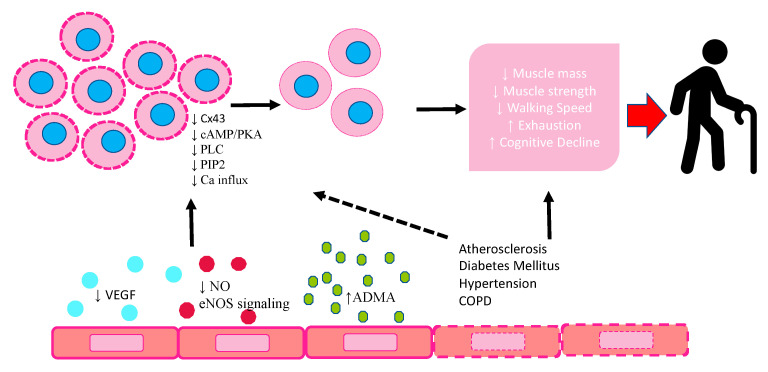
Schematic diagram highlighting the response of EPCs to endothelium dysfunction in frailty and their modification. VEGF: vascular endothelial growth factor; NO: nitric oxide; eNOS: endothelial nitric oxide synthetase; ADMA: asymmetric dimethylarginine; CX43: connexin 43; cAMP/PKA: cyclic adenosine monophosphate/ protein kinase A; PLC: phospholipase C; PIP2: phosphatidylinositol-4,5-bisphosphate.

## Data Availability

Not applicable.

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
