# Peer review of "Role of Endothelial Progenitor Cells in Frailty"

_ijms, 2023, doi:10.3390/ijms24032139_

Round 1

Reviewer 1 Report

The manuscript of Komici et al. is of interest and I commend the Authors for their nice work. My comments are minor:

- Line 36, C reactive protein is not a cytokine, but an acute phase protein. Also, in the same line please correct the synthaxys of the sentence ("are considered have been independently associated to frailty");

- Please correct alternation to alteration or impairment throughout the manuscript (it might be a mistake from autocorrect);

- It would be interesting a section describing the methods for EPCs isolation, culture, and measurement.

Author Response

Reviewer #1

The manuscript of Komici et al. is of interest and I commend the Authors for their nice work. My comments are minor:

Reply:Thank you for the revision of our manuscript for the comments and for appreciating our work.

- Line 36, C reactive protein is not a cytokine, but an acute phase protein. Also, in the same line please correct the synthaxys of the sentence ("are considered have been independently associated to frailty");

Reply: The phrase in line 36 regarding C-reactive protein was corrected and the sentence as well.

Please check page 1, lines 36-37.

- Please correct alternation to alteration or impairment throughout the manuscript (it might be a mistake from autocorrect);

Reply: As indicated 'Alternation'' was corrected to ''Alteration''.

- It would be interesting a section describing the methods for EPCs isolation, culture, and measurement.

Reply: Thank You! Following, the Reviewer's suggestion we added a section where we described the current methodology regarding EPCs isolation, culture and phenotyping, with appropriate references. Please check page 3, lines 154-170.

Reviewer 2 Report

·         The authors should add the reported studies that involved EPCs and its relation to endothelial regeneration and neovascularization.

·         English editing is highly recommended.

·         Please, re-check the punctuation, syntax, and grammar throughout the manuscript.

·         Please check all references according to the journal instructions.

Author Response

Reviewer # 2

 The authors should add the reported studies that involved EPCs and its relation to endothelial regeneration and neovascularization.

Reply:Thank you for the revision of our manuscript for the positive comments. As indicated by the Reviewer we added a paragraph where we reported studies which highlighted the role of EPCs in relationship to endothelial regeneration and neovascularization. Please check page 3 lines : 119-133.

  • English editing is highly recommended.
  • Please, re-check the punctuation, syntax, and grammar throughout the manuscript.

Reply:English language, grammar, syntax and punctation were edited in the text, and corrections were performed in the overall text.

  • Please check all references according to the journal instructions

Reply:Thank You for the remark. In the template file was not indicated a specific style, except for using square brackets for the citation in the text. In the revised version of our manuscript we used a modified Vancouver style (square brackets) for all the references.
